# Comparison of the Characteristics and Outcomes of Coronavirus Disease 2019 in Different Types of Family Infections in Taiwan

**DOI:** 10.3390/jcm9051527

**Published:** 2020-05-19

**Authors:** Shih-Feng Liu, Nai-Ying Kuo, Ho-Chang Kuo

**Affiliations:** 1Division of Pulmonary and Critical Care Medicine, Department of Internal Medicine, Kaohsiung Chang Gung Memorial Hospital, Kaohsiung 83301, Taiwan; 2Department of Respiratory Therapy, Kaohsiung Chang Gung Memorial Hospital, Taoyuan 33305, Taiwan; ellen0605@gmail.com (N.-Y.K.); erickuo48@yahoo.com.tw (H.-C.K.); 3College of Medicine, Chang Gung University, Taoyuan 33303, Taiwan; 4Department of Pediatrics, Kaohsiung Chang Gung Memorial Hospital, Kaohsiung 83301, Taiwan

**Keywords:** coronavirus disease 2019, imported family infections, domestic family infections, couple infections

## Abstract

Background: There were some family infections of coronavirus disease 2019 (COVID-19) in Taiwan to date. This study aimed to investigate the clinical characteristics and outcomes of different types of family infections with COVID-19 and to share Taiwan’s experience. Material and methods: We collected cases of family infections of COVID-19 from 21 January 2020 to 16 March 2020. The data were collected from a series of press conference contents by Taiwan’s Central Epidemic Command Center (CECC). Results: During this period, there were six family infections in Taiwan, including two couple infections, one imported family cluster infection, and three domestic family cluster infections. Compared to the former two, the starters (cases 19, 24, and 27) of domestic family cluster infections showed longer symptom-onset to diagnosis (*p* = 0.02); longer symptom-onset to quarantine or isolation (*p* = 0.01); higher first-generation reproduction number (*p* = 0.03); and more critical presentation (endotracheal tube insertion and intensive care unit (ICU) care) (*p* < 0.01). In addition, compared to the former two, the starters of the latter were older, had no history of travel, and had more underlying diseases and more mortality. There are more contacts of domestic family cluster infections, making epidemiological investigations more difficult and expensive. However, the second-generation reproduction number of the above three families was zero. Conclusion: Domestic family cluster infections of COVID-19 have different characteristics and outcomes from couple infection and imported family cluster infections in this study.

## 1. Introduction

Since the outbreak of coronavirus disease 2019 (COVID-19) in December 2019 [1,2,3], severe acute respiratory syndrome coronavirus 2 (SARS-CoV-2) has caused infections in many countries around the world [4,5]. It is more contagious than severe acute respiratory syndrome or middle east respiratory syndrome [6]. Taiwan is close to mainland China. People on both sides of the Taiwan Strait often interact [7,8]. The risk of infection in Taiwan is high [9].

The Taiwanese government has experience in dealing with SARS in the past [10,11]. Fortunately, under the leadership of the Central Epidemic Command Center (CECC), there were no large-scale COVID-19 symptomatic patients reported in Taiwan. This is due to the Taiwanese government and people’s alertness to the infection crisis. CECC’s press conference let the public know immediately the epidemic situation, so that people could raise their awareness and seek medical treatment or quarantine.

On January 21, 2020, the first case of COVID-19 pneumonia occurred in Taiwan. Later, spouses and family members became successively infected, and the number of cluster infections increased. Family infections are often sentinels of community infections. Dealing with family infections can reduce the risk of community infections. We explored the characteristics and outcomes of different types of family infections with COVID-19 and shared the experience of several family infections that have occurred in Taiwan to date to reduce future epidemics.

## 2. Materials and Methods

We collected cases of family cluster infection among COVID-19 confirmed cases from January 21, 2020 to March 16, 2020. The data were collected from a series of official information reports by the Taiwan CECC press conference. We used these data to analyze the characteristics and outcomes of COVID-19 pneumonia in different types of family infections. Case definition, specimen collection, and diagnostic tests for COVID-19 were according to Taiwan Centers for Disease Control (CDC) recommendations [12]. Clinical presentation criteria, laboratory diagnosis criteria, epidemiological criteria, and reporting requirements for COVID-19 also were regulated and published on the Taiwan CDC website [12]. Case definitions for suspected case met clinical presentation criteria but were not laboratory proven and had a history of close contact with symptomatic confirmed case(s) within 14 days prior to symptom onset. Case definitions for confirmed case met laboratory diagnosis criteria, regardless of clinical signs and symptoms [13]. The laboratory diagnosis criteria show one or more of the following: (1) Pathogen (SARS-CoV-2) isolated and identified from a clinical specimen (nasopharyngeal swab, throat swab, expectorated sputum, or lower respiratory tract aspirates). (2) Positive molecular biological testing for viral (SARS-CoV-2) RNA from a clinical specimen (nasopharyngeal swab, throat swab, expectorated sputum, or lower respiratory tract aspirates) [14].

### 2.1. Patient Informed Consents and Ethics Issue

This is a retrospective study. Patient information comes from the content of the CECC press conference and a press release from the Taiwan Centers for Disease Control and Prevention. Authors who do not know the patient’s other personal information will not be involved in privacy and personal safety. Faced with the sudden outbreak of COVID-19, we do not know where the patient is, nor can we obtain the patient’s informed consent. Given that the rights of patients have not been violated and the medical knowledge thus obtained can help more patients, ethical issues should be acceptable.

### 2.2. Study Population

As of March 16, 2020, of the confirmed cases of COVID-19 pneumonia, there were six family infections. We numbered the COVID-19 cases from 1 to X according to the order of the time of the confirmed diagnosis.

### 2.3. The First Family Couple Infection (Case 5 and Case 8)

Case 5 was a female Taiwanese businessperson in her 50s. She returned to Taiwan from Wuhan on January 20. She had fever and muscle soreness on the 25th and was diagnosed on the 27th. She was discharged from the hospital on February 23. Case 8 was the husband of Case 5. He had cough symptom on January 26 and was diagnosed on the 28th. This was the first domestic case of infection in Taiwan. He was discharged from hospital on February 27.

### 2.4. The Second Family Couple Infection (Case 9 and Case 10)

Case 9 was the second domestic case of infection in Taiwan. A woman in her 40s was infected by her husband who worked in Wuhan. Case 9 started symptoms on January 27 and was diagnosed on the 30th. She was discharged from hospital on March 3. Case 10 was the husband of Case 9, who was confirmed on January 31 and recovered and was discharged on February 13.

### 2.5. The Third Family Infection (Cases 14, 15, 17, 18)

This was the first imported family cluster infection. A couple in their 50s (Case 14 and 15) and two sons in their 20s, a family of four members, transferred from Hong Kong to Italy on 22 January and traveled from Hong Kong to Taiwan on 1 February. The couple and their eldest son had cough symptoms on 26 and 28 January, and the diagnosis was confirmed on 6 and 8 February, respectively. The youngest son was diagnosed on 9 February as the first case of “asymptomatic high virus” infection in Taiwan. He was discharged on 27 February and released from quarantine on 6 March.

### 2.6. The Fourth Family Infections (Cases 19–23)

This was the first domestic family cluster infection caused by Case 19. Case 19 was in his 60s, a taxi driver, with history of hepatitis B and diabetes, no history of going abroad. Symptoms started on 27 January and he was hospitalized on 3 February with the diagnosis of “pneumonia” at that time. He died of pneumonia combined with sepsis on 15 February and was the first death of COVID in Taiwan. The patient was diagnosed with severe pneumonia with unknown cause. He had a negative influenza test. He was notified for SARS-CoV-2 examination on 14 February and was confirmed on 16 February after death. Epidemic investigation showed that a Zhejiang businessman, who returned to Taiwan, was the source of the infection. Cases 20–23 were younger brother, mother, niece’s son-in-law, and sister of Case 19. Of the 257 close contacts, 4 (Cases 20–23) were confirmed positive, and the others were negative. All contacts were isolated and no related new confirmed cases have been identified to date. CECC announced to close the family cluster infection on 22 February.

### 2.7. The Fifth Family Infection (Cases 24–26)

This was the second domestic family cluster infection caused by Case 24. Case 24 was a woman in her 60s. She had no history of going abroad in the past two years. She had gout and hypertension. She developed fever and cough on 22 January. From 22 January to 29, she went to the clinic four times. Due to worsening symptoms and shortness of breath, she went to the hospital for emergency treatment on the evening of the 29th. She was diagnosed with pneumonia and was admitted to the hospital on the 30th. Due to her deteriorating condition, she was transferred to the intensive care unit on 10 February. Because of severe pneumonia with unknown etiology, she was notified for SARS-CoV-2 examination on 17 February and was confirmed on 19 February. Cases 25 and 26 were her granddaughter and youngest daughter. Case 26 was released from quarantine on 6 March. The index patient is currently uncertain. There were 853 contacts, of which 242 were in close contact, of which 2 were positive (Cases 25–26) and 240 were negative. All contacts were isolated and no related new confirmed cases have been identified to date. CECC announced to close the family cluster infection on 26 February.

### 2.8. The Sixth Family Infection (Cases 27–32)

This was the third domestic family cluster infection caused by Case 27. Case 27 was a male in his 80s, no recent history of going abroad. He had hypertension, diabetes, and end-stage renal disease with hemodialysis. On 6 February, he developed cough and runny nose. On the 9th, he was admitted due to pneumonia with fever. On the 16th, he was short of breath and was transferred to the intensive care unit. On the 20th, he was transferred to a negative-pressure isolation ward for suspected tuberculosis. Later, the diagnosis of COVID-19 was confirmed on the 23rd and he finally died. Cases 28–32 were the two sons, wife, grandson, and foreign caregiver of Case 27. The index patient is currently uncertain. There were 828 contacts, of which 183 remained in close contact, of which 5 were positive (Cases 28–32) and 179 were negative. All contacts were isolated and no related new confirmed cases have been identified to date. CECC announced close the family cluster infection on 27 March.

### 2.9. Statistics

Symptom-onset to diagnosis, symptom-onset to quarantine or isolation, and the first-generation reproduction number were expressed as mean/range. Independent T test was used to compare symptom-onset to diagnosis, symptom-onset to quarantine or isolation, and reproduction number between domestic family cluster infections and couple infection plus imported family cluster infections. Endotracheal tube (ET)/intensive care unit (ICU) care and mortality or not were compared using Spearman correlation test. Two-tailed *p* < 0.05 was considered statistically significant. Statistical analysis was performed using the SPSS program (version 22.0; SPSS Inc., Chicago, IL, USA).

## 3. Results

### 3.1. Demographics of Six Family Infections

There were 57 confirmed cases with COVID-19 in Taiwan as of 16 March 2020. Among the confirmed cases, there were six family infections including two couple infections, one imported family cluster infection, and three domestic family cluster infections (Figure 1).

The timeline and cases correlation of each family infection are shown in Figure 1. Case 8 was the first domestic case of COVID-19 pneumonia in Taiwan. Case 9 was transmitted by Case 10, but the diagnosis date was one day earlier than that of Case 10.

### 3.2. Characteristics and Outcome of the Starters of Three Different Types of Family Infections

Table 1 showed the characteristics and outcome of the starters of three different types of family infections. Compared to the couple infections and imported family cluster infections, the starters (Cases 19, 24, and 27) of domestic family cluster infections showed longer symptom-onset to diagnosis (21.7 vs. 5.3 days) (*p* = 0.02); longer symptom-onset to quarantine or isolation (17 vs. −3 days) (*p* = 0.01); higher first-generation reproduction number (3.7 vs. 0.67) (*p* = 0.03); and more critical presentation (endotracheal tube insertion and ICU care) (3 vs. 0) (*p* < 0.01). In addition, compared to the former two, the starters of the latter were older (60–80 years vs. 40–50 years), had no history of travel, and had more underlying diseases and more mortality. Related contact history of Case 24 and Case 27 were unknown, and the index patient has not been found to date. Case 19 was a taxi driver; after detailed medical history inquiry and epidemic investigation, the contact history of the index patient was found. Cases 19, 24, and 27 were initially diagnosed as bacterial pneumonia, were finally confirmed because of screening for pneumonia with unknown causes, and were clinically prone to delayed diagnosis. In contrast, the clinical manifestations of Cases 5, 9, and 14 had a clear history of travel and exposure and were younger, easier to diagnose, with mild severity, and eventually cured.

### 3.3. Characteristics and Outcomes of Three Different Types of COVID-19 Family Infections

Table 2 shows the characteristics and outcomes of three different types of COVID-19 family infections. There are more contacts of domestic family cluster infections, including family members, friends, medical staff and colleagues, and even some people who were not familiar with the starters. Epidemic investigations are difficult and costly. In contrast, the contacts of the couple infection and imported family cluster infection were limited to family members or close friends; the difficulty and cost of epidemic control in this case are low.

To date, the second-generation reproduction number of the above three families was zero. Three different types of family infections did not cause community infections. Of domestic family cluster infections, two infections did not find an index patient. Despite concerns about infections in the community, no new cases were identified and are still being tracked.

## 4. Discussion

Taiwan’s experience demonstrated that domestic family cluster infections have different characteristics and outcomes from couple infection and imported family cluster infections. Compared to the couple infection and imported family cluster infections, the starters (Cases 19, 24, and 27) of domestic family cluster infections showed significantly longer symptom-onset to diagnosis; longer symptom-onset to quarantine or isolation; higher first-generation reproduction number; and more critical presentation with endotracheal tube insertion and ICU care. The starters of domestic family cluster infections were older, had no history of travel, and no obvious contact history. They were not easy to diagnose early, had underlying diseases, involved many contacts, and required arduous and expensive epidemiological investigations, so they also had a higher first-generation reproduction number. The second-generation reproduction number of the above three families was zero implying that immediate responses finally controlled the spread of these family cluster COVID-19 epidemics.

Cases 19, 24, and 27 had a longer delay from the onset of symptoms to the diagnosis compared to the starters of other types of family infections. Meanwhile, Case 19 was a taxi driver with many unfamiliar customers. After the diagnosis was determined, the clues were pushed back through the communication log to find the index patient. Regarding Cases 24 and 27, detailed investigations have so far failed to find the index patients. Fortunately, medical staff and CECC remained vigilant and conducted isolation treatment before the diagnosis. Screening unknown causes of pneumonia and isolating and screening all contacts immediately after diagnosing can reduce the spread of the epidemic.

The clinical features of COVID-19 range from asymptomatic to mild to critical; most infections are not severe [15,16,17,18,19,20,21]. Fever and cough are the most common symptoms [20]. Few patients experience muscle soreness, headaches, sore throats, and runny noses [17,20]. Critical disease with respiratory failure, shock, or multi-organ dysfunction was reported in 5% of patients [21]. Acute respiratory distress syndrome (ARDS) is a major complication in critically ill patients. ARDS developed in 20% of patients after a median of 8 days, and the incidence of mechanical ventilation was 12.3% [20]. In another study of 201 hospitalized COVID-19 patients in Wuhan, 41% of patients developed ARDS, which was associated with over 65 years of age and both diabetes and hypertension [22]. Most of the fatal cases occurred in patients with advanced age or underlying medical comorbidities including cardiovascular disease, diabetes mellitus, chronic lung disease, hypertension, and cancer [20,23]. The case–fatality rate ranged from 5.8% in Wuhan to 0.7% in the rest of China [24]. People under 50 years of age with no underlying disease and no pneumonia usually show mild manifestations. For mild infections, the recovery time seems to be about two weeks, and for severe illnesses, the recovery time is three to six weeks [25].

In this study, the starters of domestic family cluster infections were at high risk of severe illness or death. The starters of couple infection and imported family clusters were mild patients. The clinical manifestations of these patients were consistent with those in previous studies [15,16,17,18,19,20,21,22,23,24,25]. All contacts should pay attention to early symptoms such as fever and cough. However, the symptoms of COVID-19 pneumonia are diverse. The order and severity of individual symptoms may vary even in the same family of infections.

### 4.1. Limitations of the Study

This study has some limitations. As the data source of the study came from the CECC press conference, we were unable to obtain individual chest X-rays, blood tests, and detailed clinical course; however, the lack of these data did not affect the purpose of this study.

### 4.2. Future Applications and Suggestions

Domestic family cluster infections have different characteristics and outcomes from couple infection and imported family cluster infections. Domestic family cluster infections require more medical or public resources than couples and imported cluster family infections. Due to the lack of a clear history of touch, occupation, contact, cluster (TOCC), the starters of domestic family cluster infections are not easy to diagnose early, and there is a risk of community infection. Strengthening screening for unexplained pneumonia and a detailed history of TOCC can help detect these cases early. Early identification and isolation can prevent community infections as much as possible.

## Figures and Tables

**Figure 1 jcm-09-01527-f001:**
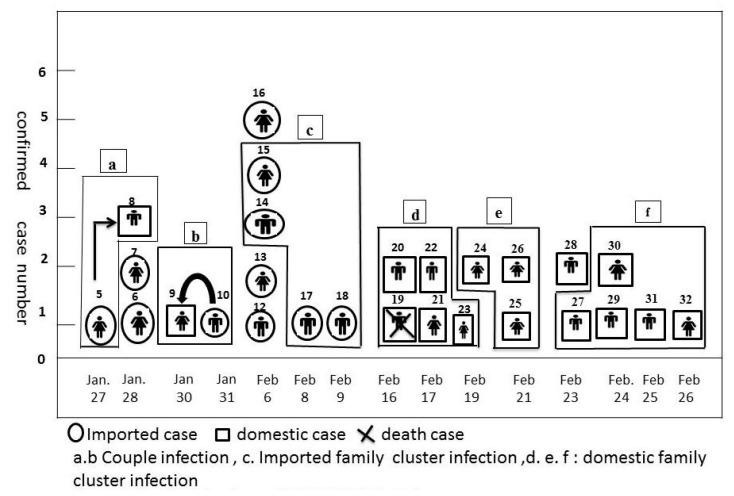
Family infections of severe acute respiratory syndrome coronavirus 2 (SARS-CoV-2) in Taiwan.

**Table 1 jcm-09-01527-t001:** Comparison of the characteristics and outcome of starters of different types of family infections with coronavirus disease 2019 (COVID-19) in Taiwan.

	Couple Infection	Imported Family Cluster Infection	Domestic Family Cluster Infection
Starter case No	Case 5	Case 14	Case 19
	Case 9		Case 24
			Case 27
Age/gender	Case 5: 50 s/F	Case 14: 50 s/M	Case 19: 60 s/M
			Case 24: 60 s/F
	Case 10: 40 s/M		Case 27: 80 s/M
Travel history	Yes	Yes	No
Touch history	Obvious	Obvious	Case 19: not obvious *
			Case 24: unknown ^
			Case 27: unknown ^
Underling disease	No	No	Case 19: DM, B hepatitis
			Case 24: HTN, gout
			Case 27: HTN, DM
Disease severity	Mild	Mild	Severe and critical
			Case 19: ARF, shock, ICU care
			Case 24: ARF, ICU care
			Case 27: ARF, ICU care
Initial diagnosis	Suspected COVID-19	Suspected COVID-19	No suspected COVID-19
Incubation period	Case 5: 5 days	4 days	Case 19: 6 days
			Case 24: unknown ^
	Case 9: 6 days		Case 27: unknown ^
Symptom-onset to diagnosis Mean/range	Case 5: 2 days	11 days	Case 19: 20 days
	Case 9: 3 days		Case 24: 28 days
			Case 27: 17 days
	5.3	/2–11 days	21.7/17–28 days ** (*p* = 0.02)
Symptom-onset to quarantine or isolation Mean/range	Case 5: −2 days ^&^	−11 days ^&^	Case 19: 18 days
	Case 9: 4 days		Case 24: 19 days
			Case 27: 14 days
	−3	/−11–4	17/14–19 days ** (*p* = 0.01)
Diagnosis to discharge	Case 5: 33 days	35 days	Case 19: −1 day ^@^ (mortality)
	Case 9: 37 days		Case 24: in admission
			Case 27: 26 days (mortality)
Reproduction			
1st generation	Case 5: *n* = 1	Case14: *n* = 0 ^#^	Case 19: *n* = 4
	Case 9: *n* = 1		Case 24: *n* = 2
			Case 27: *n* = 5
Mean/range	0.67	/0–1	3.7/2–5 ** (*p* = 0.03)
2nd generation	*n* = 0	*n* = 0	*n* = 0
Cause of screening for COVID-19	Travel history	Travel history	Severe pneumonia with unknown cause
Clinical course on ET/ICU care outcome	No		All: yes ** (*p* < 0.01)
	Case 5: recovery	No	Case 19: mortality
	Case 9: recovery	Case14: recovery	Case 24: in ICU treatment
			Case 27: mortality

** *p* < 0.05 (domestic family cluster infection vs. couple infection + imported family cluster infection). * Epidemic investigation showed that a Zhejiang businessman, who returned to Taiwan and took Case 19’s taxi and was confirmed as the index patient. ^ Infectious source was not determined to date. ^#^ A family four members transferred from Hong Kong to Italy on January 22 and traveled from Hong Kong to Taiwan on February 1. ^@^ Case 19 died on February 15th and was confirmed on February 16. ^&^ Case 5 and Case 14 were isolated since they arrived in Taiwan from aboard. Abbreviations: COVID-19: coronavirus disease 2019 M: male; F: female; DM: diabetic mellitus; HTN: hypertension; ARF: acute respiratory failure; ICU: intensive care unit; ET: endotracheal tube. 50s/F: 50–60y/o/F (female in her 50s).

**Table 2 jcm-09-01527-t002:** Comparison of the characteristics and outcome of three types of family infections with COVID-19 in Taiwan.

	Couple Infection	Imported Family Cluster Infection	Domestic Family Cluster Infection
Case No.	Case 5 and Case 8	Case 14–18	Case 19–23
	Case 9 and Case 10		Case 24–26
			Case 27–32
All contacts	Family members	Family members	Case 19: 257 persons
			Case 24: 853 persons
			Case 27: 828 persons
Contacts outcome	Case 8 positive	Case 15–18 positive	All contacts negative except above infected family members
Community infection	No	No	No *
Cost of epidemiological investigation	Lower	Lower	Higher ^#^
Difficulty of epidemiological investigation	Lower	Lower	Higher ^#^
Risk of community infection	Lower	Lower	Higher ^#^

* All contacts of the three domestic clusters family infection were isolated and no related new confirmed cases were identified to date. ^#^ Compared with couple infection and imported family cluster infection.

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
