# Peer review of "Comparison of the Characteristics and Outcomes of Coronavirus Disease 2019 in Different Types of Family Infections in Taiwan"

_jcm, 2020, doi:10.3390/jcm9051527_

Round 1

Reviewer 1 Report

Summary

In this article, the authors describe clinical characteristics and outcomes of COVID-19 infections in different family units from Taiwan.

Broad Comments

  1. L40-44: Clarification is needed in this paragraph. Unless, everyone was tested at that time, then you cannot say that there was no large-scale COVID-19 community infection. What can be said is that there was no large-scale COVID-19 symptomatic patients reported. On L42, this statement should be suggested unless there is data to back up this statement.
  2. Was there any information available on the method (PCR or other method?) used to test and confirm SARS-Cov-2 infection in patients?
  3. How was a contact defined? For example, on L107, case 24 had 853 contacts.
  4. Most of the analysis is focused on the “starter” cases of which there were 6 in total. Due to the small number of patients, conclusions should be made with caution.

Specific Comments

  1. L16: Change “OVID-19” to “COVID-19”.
  2. L27-288: This conclusion statement is too strong considering the small number of cases that were investigated.
  3. L44: Consider removing this statement, “to appease the hearts of people, and cooperates with the entire people”.
  4. L62-63: Delete, “I believe that everyone will have the same consensus”.
  5. L69, L76, L100, L102: Change “He” to “She”.
  6. L111: Change “fifth” to “sixth”.
  7. L138 (Figure 1): What does the y-axis, confirmed case number mean? What do the people outside of the broader boxes signify (e.g., #6 and #7)? More explanation is needed for Figure 1.
  8. L155 (Table 1): Define abbreviation such as DM, HTN, ARF, and others used in Table 1.
  9. L174 (Table 2): What is considered “low” vs “high” in terms of cost of epidemiological investigation?

Author Response

Thanks for the reviewers ’valuable opinions. We response them point-by-point and revise them in red color in the manuscript. The line numbers will be different from those of the original version.

Reviewer1

Summary

In this article, the authors describe clinical characteristics and outcomes of COVID-19 infections in different family units from Taiwan.

Broad Comments

1.L40-44: Clarification is needed in this paragraph. Unless, everyone was tested at that time, then you cannot say that there was no large-scale COVID-19 community infection. What can be said is that there was no large-scale COVID-19 symptomatic patients reported. On L42, this statement should be suggested unless there is data to back up this statement.

R: The current practice in Taiwan is still doing as mentioned in the method (we add a paragraph in M&M), without comprehensive screening. So we revise” Fortunately, under the leadership of Central Epidemic Command Center (CECC), there is currently no large-scale COVID-19 community infection in Taiwan. Change to: Fortunately, under the leadership of Central Epidemic Command Center (CECC), there were no large-scale COVID-19 symptomatic patients reported in Taiwan (line40, 41).

  1. Was there any information available on the method (PCR or other method?) used to test and confirm SARS-Cov-2 infection in patients?

R: We add a paragraph in Materials and Methods (line 55-66) and add some reference [12-14] (line 284-289). The Diagnostic detection of 2019-nCoV use real-time RT-PCR test.

Case definition, specimen collection, and diagnostic tests for COVID-19 are according to Taiwan CDC recommendations [12]. Clinical presentation criteria, laboratory diagnosis criteria, epidemiological criteria, and reporting requirements for COVID-19 also were regulated and published on the Taiwan CDC website [12]. Case definitions for suspected case meet clinical presentation criteria but not laboratory proven plus history of close contact with symptomatic confirmed case(s) within 14 days prior to symptom onset. Case definitions for confirmed case meet laboratory diagnosis criteria, regardless of clinical signs and symptoms [13]. The laboratory diagnosis criteria show one or more of the following: (1) Pathogen (SARS-CoV-2) isolated and identified from a clinical specimen (nasopharyngeal swab, throat swab, expectorated sputum, or lower respiratory tract aspirates). (2) Positive molecular biological testing for viral (SARS-CoV-2) RNA from a clinical specimen (nasopharyngeal swab, throat swab, expectorated sputum, or lower respiratory tract aspirates) [14]. (line 55-66)

  1. How was a contact defined? For example, on L107, case 24 had 853 contacts.

R: CDC press release as below: There are 853 contacts, of which 242 are in close contact, of which 2 are positive (cases 25-26) and 240 are negative.

CECC did not clearly explain the definition of linkage. Generally speaking, close contact is required to do throat swab, sputum or serum for PCR examination. Personal understanding is as follows: close contact should include family members, as well as friends and work colleagues who often contact these members. The other 616 people should be in contact during this period by history taking, but they cannot be contacted regularly.

  1. Most of the analysis is focused on the “starter” cases of which there were 6 in total. Due to the small number of patients, conclusions should be made with caution.

R: Thanks for your suggestion. We revise it in Abstract conclusion and 4.2. Future Applications and Suggestions. (line 28-29;and delete in line 241)

Specific Comments

5.L16: Change “OVID-19” to “COVID-19”.

R: Change “OVID-19” to “COVID-19” (line 16)

6.L27-288: This conclusion statement is too strong considering the small number of cases that were investigated.

R: Delete Taiwan CECC’s measures effectively reduce the spread of these family cluster COVID-19 epidemic.

Conclusion revision: Domestic family cluster infections of COVID-19 have different characteristics and outcomes from couple infection and imported family cluster infections in this study. (line 28-29)

7.L44: Consider removing this statement, “to appease the hearts of people, and cooperates with the entire people”.

R: We delete this sentence

We revise it as below: CECC’s press conference let the public know immediately the epidemic situation, so that people could raise their awareness and seek medical  treatment or quarantine. (line 42-44)

8.L62-63: Delete, “I believe that everyone will have the same consensus”.

R: We delete this sentence. Given that the rights of patients have not been violated and the medical knowledge thus obtained can help more patients, ethical issues should be acceptable (line 73-74)

9.L69, L76, L100, L102: Change “He” to “She”.

R: Change He to She in L81, L88, L113, L115

10.L111: Change “fifth” to “sixth”.

R: Change “fifth” to “sixth” in line 124.

11.L138 (Figure 1): What does the y-axis, confirmed case number mean? What do the people outside of the broader boxes signify (e.g., #6 and #7)? More explanation is needed for Figure 1.

R: confirmed case number means the numbers of every day, ex one COVID-19 case on Jan 27, three cases on Jan 28

Case 6 and 7 are not family and do not belong to the members of family cluster infection (a, b c, d, e, f), so they were listed outside of the broader boxes.

Case X: We numbered the COVID-19 cases from 1 to X according to the order of the time of the confirmed diagnosis (we add it in the 1.2 study population line 77-78)

12.L155 (Table 1): Define abbreviation such as DM, HTN, ARF, and others used in Table 1.

R: We add definition of abbreviation such as DM, HTN, ARF, and others used in Table 1. Abbreviations: COVID-19: coronavirus disease 2019 M: male; F: female; DM: diabetic mellitus; HTN: hypertension; ARF: acute respiratory failure; ICU: intensive care unit; ET: endotracheal tube (Table 1 footnote) (line 177-178)

13.L174 (Table 2): What is considered “low” vs “high” in terms of cost of epidemiological investigation?

R: We change it lower vs higher and add footnote in table 2

#compared with couple infection and imported family cluster infection (line 193-194)

Reviewer 2 Report

The manuscript may had been interesting; however, lack of clear relation between results and conclusions, lack of description of measures and the fact of  the outbreak of Coronavirus pandemic in Taiwan from middle of March to middle of April 2020 greatly reduce this reviewer’s enthusiasm about this study.

Major concerns stem from the:

1) The authors concluded that Taiwan CECC’s measures effectively reduce the spread of this family cluster COVID-19 epidemic. Unfortunately, there was lack of the clear description of Taiwan CECC’s measures and how did it work.

2) One of the major results in this study is that “compared to the former two, the starters (cases 19, 24, and 27) of domestic family cluster infections showed longer symptom-onset to diagnosis (P=0.02)". Isn't this difference because of possible “Taiwan CECC‘s measures” so that infected people with a history of going abroad will get a priority diagnosis after they have symptoms? What if the starter of the domestic family also get priority diagnosis so that the family members can be protected? Remember the other major result of “more the first generation of reproduction number”.

3) low statistical significance of the low number of the cases for each group since the clinical parameters/info like “the days of symptom-onset to diagnosis” and “symptom-onset to diagnosis” may also be related to the individual constitution of each infected person.

Author Response

Thanks for the reviewers ’valuable opinions. We response them point-to-point and revise them in red color in the manuscript. The line numbers will be different from those of the original version.

Reviewer 2

The manuscript may had been interesting; however, lack of clear relation between results and conclusions, lack of description of measures and the fact of the outbreak of Coronavirus pandemic in Taiwan from middle of March to middle of April 2020 greatly reduce this reviewer’s enthusiasm about this study.

Major concerns stem from the:

1) The authors concluded that Taiwan CECC’s measures effectively reduce the spread of this family cluster COVID-19 epidemic. Unfortunately, there was lack of the clear description of Taiwan CECC’s measures and how did it work.

R: We delete the sentence in Abstract conclusion and 4.2. Future Applications and Suggestions

2) One of the major results in this study is that “compared to the former two, the starters (cases 19, 24, and 27) of domestic family cluster infections showed longer symptom-onset to diagnosis (P=0.02)". Isn't this difference because of possible “Taiwan CECC‘s measures” so that infected people with a history of going abroad will get a priority diagnosis after they have symptoms? What if the starter of the domestic family also get priority diagnosis so that the family members can be protected? Remember the other major result of “more the first generation of reproduction number”.

R: Thanks for your opinion. We add a paragraph in M&M (line 55-66). Case definition, specimen collection, and diagnostic tests for COVID-19 are according to Taiwan CDC recommendations [12]. Clinical presentation criteria, laboratory diagnosis criteria, epidemiological criteria, and reporting requirements for COVID-19 also were regulated and published on the Taiwan CDC website [12]. Case definitions for suspected case meet clinical presentation criteria but not laboratory proven plus history of close contact with symptomatic confirmed case(s) within 14 days prior to symptom onset. Case definitions for confirmed case meet laboratory diagnosis criteria, regardless of clinical signs and symptoms [13]. The laboratory diagnosis criteria show one or more of the following: (1) Pathogen (SARS-CoV-2) isolated and identified from a clinical specimen (nasopharyngeal swab, throat swab, expectorated sputum, or lower respiratory tract aspirates). (2) Positive molecular biological testing for viral (SARS-CoV-2) RNA from a clinical specimen (nasopharyngeal swab, throat swab, expectorated sputum, or lower respiratory tract aspirates) [14]. The current practice in Taiwan is still doing as mentioned in the method, without comprehensive screening. Of course, Contacts of the confirmed cases will be monitored more rigorously. We believe as your opinion “if the starter of the domestic family also get priority diagnosis so that the family members can be protected” and “longer symptom-onset to diagnosis”. At the beginning, the resources were also limited. It is difficult for us to comprehensively examine patients who have no history of contact and clinical history.

In addition, we add the sentence about the first generation of reproduction number in line 205-207. so also have more the first generation of reproduction number. The 2nd generation of reproduction number of the above three family was zero implies that immediate responses finally control the spread of these family cluster COVID-19 epidemics.(line204-207)

3) low statistical significance of the low number of the cases for each group since the clinical parameters/info like “the days of symptom-onset to diagnosis” and “symptom-onset to diagnosis” may also be related to the individual constitution of each infected person

R: Thanks for your opinion. When we wrote this article, it was true that the cases of family cluster infections in Taiwan were these people by official records, so only these people can be analyzed. The statistical efficiency may not be very high. Not all people in Taiwan do the COVID-19 PCR test. We have added this paragraph in the M&M, so it is true that some cases may not be detected early for especially these starters of domestics family cluster infection. Just like your opinion, some of them may be delayed in diagnosis and cause family infection. Also as your valuable opinion: “symptom-onset to diagnosis ”may also be related to the individual constitution of each infected person. This is where we have to review and improve.

Round 2

Reviewer 1 Report

The authors have addressed previous comments.

Reviewer 2 Report

The author's reply removed the reviewer's most concerns.